# Benzoxazine–Purine Hybrids as Antiproliferative Agents: Rational Design and Divergent Mechanisms of Action

**DOI:** 10.3390/pharmaceutics17101260

**Published:** 2025-09-26

**Authors:** Houria Boulaiz, Yaiza Jiménez-Martínez, Francisco Franco-Montalbán, Jesús Peña-Martín, Ana Conejo-García, M. Dora Carrión

**Affiliations:** 1Department of Human Anatomy and Embryology, Faculty of Medicine, University of Granada, 18016 Granada, Spain; hboulaiz@ugr.es; 2Biosanitary Institute of Granada (ibs.GRANADA), SAS-University of Granada, Avenida de Madrid, 15, 18012 Granada, Spain; yaijmartinez@correo.ugr.es (Y.J.-M.); jespmar@ugr.es (J.P.-M.); 3Biopathology and Regenerative Medicine Institute (IBIMER), Centre for Biomedical Research, University of Granada, 18016 Granada, Spain; 4Excellence Research Unit of Chemistry Applied to Biomedicine and the Environment, Department of Medicinal and Organic Chemistry, Faculty of Pharmacy, University of Granada, Campus Cartuja s/n, 18071 Granada, Spain; ffranco@ugr.es

**Keywords:** benzoxazines, purines, kinases inhibition, anticancer activity, apoptosis and pyroptosis

## Abstract

**Background/Objectives:** Targeted cancer therapies increasingly rely on modulating specific cell death pathways and kinase signaling. Due to their structural versatility and potential to induce mechanistically distinct cytotoxic responses, benzoxazine–purine hybrids represent a promising scaffold for anticancer drug development. The objective of this study was to design and evaluate novel benzoxazine–purine derivatives for their antiproliferative activity and elucidate their underlying mechanisms of action. **Methods:** A series of benzoxazine–purine compounds was synthesized via a modular and efficient approach. The synthetic route involved a one-pot cyclization of substituted 2-aminophenols with epichlorohydrin, followed by tosylation and subsequent Mitsunobu coupling with halogenated purines. Their antiproliferative activity was assessed in MCF-7 (breast) and HCT-116 (colon) cancer cell lines using MTT assays. Selected compounds were evaluated further for kinase inhibition, effects on the cell cycle, membrane integrity (Annexin V/PI staining), ultrastructural changes (SEM), and caspase activation (Western blot). In silico ADMET profiling was also performed. **Results:** Compounds **9** and **12** exhibited the most potent antiproliferative activity, with low micromolar IC_50_ values. Compound **12** showed dual HER2/JNK1 kinase inhibition and induced caspase-8-dependent pyroptosis-like cell death, characterized by membrane rupture and inflammatory features. In contrast, compound **8** lacked kinase inhibition and promoted S-phase arrest with apoptotic-like morphology. Both compounds demonstrated favorable physicochemical and ADMET profiles, including high intestinal absorption and an absence of mutagenicity. **Conclusions:** The rational design of benzoxazine–purine hybrids resulted in the discovery of compounds with distinct mechanisms of action. Compound **12** induces inflammatory cell death by modulating kinases, while compound **9** acts through a kinase-independent apoptotic pathway. These results underscore the therapeutic potential of scaffold-based diversification for developing targeted anticancer agents.

## 1. Introduction

Cancer is the leading cause of death worldwide and a major obstacle to improving life expectancy in all countries. The global burden of cancer, in terms of both incidence and mortality, is increasing rapidly [1]. Breast cancer stands as the most frequently diagnosed malignancy among women [2,3], while colorectal cancer has become increasingly prevalent in recent years [4]. Consequently, research and development of new, effective and safe anticancer drugs remains a major focus of today’s society [5].

Protein kinases are central regulators of cancer cell signaling and have emerged as critical therapeutic targets. Among them, HER2, a receptor tyrosine kinase of the ErbB family, plays a key role in promoting tumorigenesis through pathways controlling proliferation, survival, and differentiation. HER2 overexpression is notably associated with aggressive forms of breast cancer and correlates with poor clinical outcomes [6]. Similarly, the c-Jun N-terminal kinase (JNK) pathway, a component of the mitogen-activated protein kinase (MAPK) family, is involved in cellular stress responses and has been linked to both apoptotic and non-apoptotic forms of cell death. Recent studies have highlighted JNK1′s contribution to pyroptosis in certain cancer cell types, suggesting its potential as a therapeutic target in inflammation-associated tumor progression [7].

A deeper understanding of cell death mechanisms triggered by anticancer agents is crucial for overcoming resistance and minimizing toxicity. While apoptosis remains the most widely studied form of programmed cell death, increasing attention has turned to alternative pathways, such as pyroptosis, which may be exploited to bypass apoptosis resistance in cancer cells. Pyroptosis is characterized by inflammatory membrane rupture and is typically mediated by the gasdermin family following caspase activation. Notably, several tumors exhibit defective apoptotic machinery due to impaired death receptor signaling, altered expression of Bcl-2 family proteins, or reduced caspase activity [8]. Thus, targeting caspases, not only for apoptosis but also for their emerging role in inflammatory cell death, has become a promising strategy in anticancer drug development [9].

Purines are well-known for their significant role as antitumor agents due to their ability to interfere with DNA and RNA synthesis, leading to the inhibition of cancer cell proliferation [10,11]. Our research group has been actively involved in developing novel compounds based on purine structures, which have shown remarkable potential in anticancer therapy [12,13,14].

Benzoxazines are considered valuable scaffolds in medicinal chemistry due to their versatile structure, which allows multiple modifications and has shown significant antitumor activity in various cancer cell lines [15].

Recent advances in medicinal chemistry have highlighted the therapeutic potential of hybrid molecules that combine purine cores and benzoxazine moieties. These hybrid structures combine the biological relevance of purines with the structural adaptability of benzoxazines, resulting in compounds with promising pharmacological profiles [16,17]. The strategic fusion of these two heterocyclic systems is a promising approach for developing multifunctional agents for use in anticancer and antiviral therapy.

We have previously reported compound **1** with an ethyl linker between the benzoxazine system and the purine (Figure 1), which exhibited activity against MCF-7 breast cancer cells (IC_50_ of 13 μM) and the HCT-116 colon cancer cell line (IC_50_ 7.06 μM) [18]. Building on this work, we recently designed a series of benzoxazine derivatives incorporating different substituents in the aromatic ring while preserving the ethyl linker to the purine ring. Among these, derivative **2** showed particularly promising antiproliferative properties, with IC_50_ values of 2.27 and 4.44 μM against MCF-7 and HCT-116 cell lines, respectively. This compound exerted its cytotoxic activity by disrupting cell membrane permeability, likely triggering both inflammatory and non-inflammatory cell death mechanisms (Figure 1) [19].

In an effort to find new and potent antiproliferative agents and to gain further insight into the mechanism of action by which such derivatives induce cell death, we have designed a new series of substituted benzoxazines. These derivatives have electron-withdrawing groups (**3**–**10**) and electron-donating groups (**11**–**14**) linked by a methyl to a substituted purine system (Figure 1). We have eliminated a rotatable bond in these compounds, a pharmacomodulation strategy known as truncation. This modification is expected to increase the stability and potency of the compounds by reducing conformational flexibility. In addition, the synthesis of these new derivatives (**3**–**14**) offers the advantage of a shorter synthetic route, making it a more efficient method.

## 2. Material and Methods

### 2.1. Chemistry

#### 2.1.1. General

All chemical reagents and solvents were sourced from Sigma-Aldrich (St. Louis, MO, USA), Thermo Fisher Scientific (Waltham, MA, USA), VWR International Ltd. (Leicestershire, UK), or Fluorochem (Derbyshire, UK). Thin-layer chromatography (TLC) analyses were conducted on Merck Kieselgel 60 F254 aluminum-backed plates (Darmstadt, Germany) and visualized under UV light at 254 nm. Solvent removal was performed under reduced pressure using a Büchi rotary evaporator (Flawil, Switzerland) equipped with a Vacuubrand CVCII pressure controller (Wertheim, Germany). Compound purification was achieved via preparative TLC or flash column chromatography employing silica gel 60 (particle size 0.040 mm–0.063 mm; 230 mesh–440 mesh ASTM). Melting points were measured using open capillary tubes on a Stuart SMP3 melting point apparatus (Staffordshire, UK). Microwave-assisted reactions on a small scale were conducted using an Initiator 2.0 single-mode microwave reactor, which delivers controlled irradiation at 2.450 GHz (Biotage AB, Uppsala, Sweden).

NMR spectra were acquired at ambient temperature on a Varian DirectDrive spectrometer (Palo Alto, CA, USA) operating at 500 or 400 MHz for ^1^H NMR, and at 126 or 101 MHz for ^13^C NMR. Chemical shifts (δ) are reported in parts per million (ppm) relative to the residual solvent peak. Signal multiplicities are designated as s (singlet), d (doublet), dd (doublet of doublets), t (triplet), and m (multiplet); coupling constants (*J*) are given in hertz (Hz). All NMR samples were prepared in CDCl_3_.

High-resolution mass spectrometry data were obtained using a Bruker compact QTOF system with electrospray ionization (ESI) operated in either positive or negative ion mode (Bruker Daltonik GmbH, Bremen, Germany), depending on compound structure. Instrument settings included: nebulizer pressure of 1.8 bar, capillary voltage of ±4500 V, mass scan range 100 *m*/*z*–2000 *m*/*z*, dry gas flow of 9.0 L/min, and a dry heater temperature of 220 °C.

#### 2.1.2. Synthesis of (6 or 7 Substituted-3,4 Dihydro-2H-benzo[b][1,4]oxazin-2-yl)methanol (Compounds **18**–**20**)

The corresponding substituted 2-aminophenol (**15**, **16**, **17**) 6 mmol was dissolved in 30 mL of H_2_O in presence of 8.4 mmol of NaOH. Then, 7.2 mmol of epichlorohydrin was added dropwise to this solution. After addition, it was left to stir at room temperature for one hour, following the course of the reaction by thin layer chromatography (TLC). The reaction mixture was extracted with dichloromethane (DCM), the organic phase was dried over anhydrous sodium sulfate and concentrated under vacuum. The crude obtained was purified by silica gel column chromatography using a solvent mixture of ethyl acetate and hexane (EtOAc/Hex, 1:2).

#### 2.1.3. Synthesis of (6 or 7 Substituted-4-tosyl-3,4-dihydro-2H-benzo[b][1,4]oxazin-2-yl)methanol (Compounds **21**–**23**)

The corresponding alcohol (**18**, **19**, **20**) 2.048 mmol was dissolved in 20 mL of anhydrous dichloromethane under argon atmosphere. After that, 246 µL of pyridine was added and cooled the reaction in an ice bath. Then, *p*-toluensulfonyl chloride (2.048 mmol) was added. The ice bath was removed, and the reaction was stirred at room temperature for 24 h. After this time, the reaction was extracted with dichloromethane and the organic phase was washed with 5 mL 1N HCl and brine. The organic phase was dried over anhydrous sodium sulfate and concentrated under vacuum. The crude product obtained was purified by flash chromatography (EtOAc/hexane 1:1).

#### 2.1.4. Synthesis of Substituted 2-((6-halo or 2,6-dihalo-9H-purin-9-yl)methyl)-4-tosyl-3,4-dihydro-2H-benzo[b][1,4]oxazine Derivatives (Compounds **3**–**5**, **7**–**9**, **11**–**13**)

Under argon atmosphere the corresponding substituted purine (6-chloro, 2,6-dichloro or 6-bromopurine commercially available from Sigma-Aldrich) (0.395 mmol), triphenylphosphine (0.73 mmol) and the appropriate tosylated derivative (**21**, **22**, **23**) (0.36 mmol) were dissolved in anhydrous tetrahydrofuran (4 mL). The reaction mixture was cooled to −20 °C and DIAD (0.73 mmol) was added. Then, the mixture was stirred at room temperature for 48 h. After this time, the reaction was concentrated under vacuum and the crude was purified by flash chromatography using EtOAc/hexane 1:1.

#### 2.1.5. Synthesis of Substituted 4-tosyl-2-((6-(trifluoromethyl)-9H-purin-9-yl)methyl)-3,4-dihydro-2H-benzo[b][1,4]oxazine Derivatives (Compounds **6**, **10**, **14**)

The 6-bromo derivative **5**, **9** or **13** (0.3 mmol) was added to a solution containing MFSDA (0.52 mmol), CuI (0.36 mmol) and HMPA (0.37 mmol) in anhydrous DMF (4 mL). The mixture was microwaved at 150 °C for 15 min. EtOAC was added and the mixture was washed with saturated NH_4_Cl, NaHCO_3_, H_2_O and brine. The organic phase was dried over anhydrous Na_2_SO_4_, filtered, and evaporated under vacuum. The crude product obtained was purified by flash chromatography (using EtOAc/hexane 9:1).

The compound purity analysis characterizations of the intermediates (**18**–**23**) and target compounds (**3**–**14**) by NMR and HRMS is included in the Appendix A.

### 2.2. Biology

#### 2.2.1. Cell Culture Conditions

The HCT-116 (colorectal carcinoma) and MCF-7 (breast adenocarcinoma) human cancer cell lines were supplied by the Cell Bank of the University of Granada. Cells were maintained in Dulbecco’s Modified Eagle Medium (DMEM, Sigma-Aldrich, St. Louis, MO, USA) enriched with 10% heat-inactivated fetal bovine serum (FBS) and 1% penicillin–streptomycin (P/S; Sigma-Aldrich). Cultures were incubated at 37 °C under a humidified atmosphere with 5% CO_2_. Only early-passage cells (within six months of thawing) were used for experiments, and routine mycoplasma testing was performed to ensure the absence of contamination.

#### 2.2.2. Preparation and Application of Compounds

Test compounds (**3**–**14**) were initially dissolved in dimethyl sulfoxide (DMSO) to prepare stock solutions, which were aliquoted and stored at −20 °C. For each assay, fresh dilutions were prepared in complete culture medium to achieve the desired working concentrations. The final DMSO concentration in cell treatments did not exceed 0.1% (*v*/*v*), a level previously confirmed to have no cytotoxic effects. Control samples were treated with equivalent volumes of DMSO to account for any solvent-related influence.

#### 2.2.3. In Vitro Cell Viability Assay

To determine compound cytotoxicity, the MTT assay was employed using Thiazolyl Blue Tetrazolium Bromide (Sigma-Aldrich). Cells (2500 per well) were seeded into 96-well plates and incubated for 24 h to allow adherence. Following this, cells were treated with different compound concentrations for 72 h. After incubation, MTT was added and incubated for an additional 3 h. Formazan crystals were solubilized using ≥ 99.5% DMSO (Sigma-Aldrich, Saint-Quentin-Fallavier, France). Absorbance was measured at 570 nm using a Titertek Multiscan microplate reader (Flow Laboratories, Irvine, CA, USA). IC_50_ values and dose–response curves were derived using the four-parameter logistic model with the drc package in R (version 3.0-1).

#### 2.2.4. Kinase Inhibition Assays

The inhibitory profiles of compounds **5**, **8**, **9**, **10**, **11**, and **12** were evaluated against a panel of five protein kinases, as shown in Appendix A. Each compound was tested at two concentrations (5 × 10^−5^ M and 5 × 10^−6^ M) in a single replicate for each kinase assay. Residual enzymatic activity was then determined. The Appendix A provides comprehensive experimental procedures for the kinome analysis.

#### 2.2.5. Scanning Electron Microscopy

Cell morphology was evaluated via scanning electron microscopy (SEM). HCT-116 and MCF-7 cells were seeded onto sterile glass coverslips (13 mm diameter) in 24-well plates and cultured for 24 h before treatment. Cells were then incubated for 12 h with compounds 9 or 12 (50 μM). After treatment, cells were washed in cold PBS, fixed, and processed for SEM according to established protocols. Imaging was conducted using a Hitachi S-800 microscope (Hitachi, Tokyo, Japan).

#### 2.2.6. Cell Cycle Distribution Analysis

Cell cycle progression was evaluated by propidium iodide (PI) staining and flow cytometry. HCT-116 and MCF-7 cells were exposed to 50 μM of **9** or **12** for 12 h. Following treatment, cells were harvested, washed twice in PBS, and fixed with 70% cold ethanol at 4 °C for 30 min. After fixation, samples were incubated with PI/RNase staining buffer (BD Pharmingen™, Cat. No. 550825 according to the manufacturer’s instructions. DNA content analysis was carried out using a FACSCalibur flow cytometer (BD Biosciences, Franklin Lakes, NJ, USA), and the distribution of cells across the G_0_/G_1_, S, and G_2_/M phases was determined using standard analytical software.

#### 2.2.7. Annexin V-Based Apoptosis Assay

Apoptosis was assessed using a fluorometric Annexin V-FITC assay kit (eBioscience, San Diego, CA, USA). Cells were plated in 6-well plates and treated with 50 μM of either compound 9 or 12 for 12 h. Staining was carried out as per manufacturer’s instructions. Flow cytometry analysis was conducted immediately using a FACSAria III (BD Biosciences) at the Scientific Instrumentation Centre, University of Granada.

#### 2.2.8. Western Blot Assay

Protein expression was analyzed by Western blot after treating cells with 50 μM of the test compound for 4 h. Cells were lysed in RIPA buffer (Santa Cruz Biotechnology, Heidelberg, Germany). Proteins were separated via 5–15% SDS-PAGE and transferred onto PVDF membranes. Membranes were blocked in 5% milk with 0.1% Tween-20 in PBS for 1 h and incubated overnight with primary antibodies against caspase-8 (sc-81656), caspase-3 (sc-56053), and caspase-1 (sc-56036). After overnight incubation and washing, membranes were incubated with HRP-conjugated secondary antibodies (sc-2005) and developed using SuperSignal™ West Dura substrate (Thermo Scientific, Waltham, MA, USA). Chemiluminescence was detected using the ChemiDoc MP system (BioRad, Hercules, CA, USA) at the Scientific Instrumental Center (University of Granada), and band intensities were quantified with ImageJ software (version 1.53a).

#### 2.2.9. Statistical Analysis

All experimental data were obtained from three independent replicates. Statistical significance was assessed using Student’s *t*-test. Results are presented as mean ± SD. Differences were considered significant at *p* < 0.05 (*), *p* < 0.01 (**), and *p* < 0.001 (**).

### 2.3. Docking Studies

We conducted molecular docking studies using AutoDock 4.2.6 (AD4) [20], targeting the human epidermal growth factor receptor 2 (hHER2; PDB ID: 3RCD, subunit B) and the human mitogen-activated protein kinase 8 (hJNK1; PDB ID: 4AWI). Both enantiomeric forms of the ligands were constructed in Avogadro (version 1.2.0) [21] and geometrically optimized at the HF/6-31G(d,p) level using Gaussian software (version 0.9) [22]. The ligand files were processed for docking using the prepare_ligand4.py utility provided with MGLTools 1.5.4. [20]. The protein structures were prepared with PDB2PQR (version 3.6.2) [23] by removing water molecules and co-crystallized ligands and by assigning charges and nonpolar hydrogens at a pH of 7.0. The resulting structures were formatted for docking using the prepare_receptor4.py script. Docking grids were centered at the orthosteric sites of the respective targets, defined with dimensions of 75 Å × 80 Å × 70 Å, and spaced 0.375 Å apart.

Docking simulations employed the Lamarckian genetic algorithm with 100 runs of the genetic algorithm, 2.5 million energy evaluations, and a population size of 150 by default. Additionally, a local search using the Solis and Wets method was executed with 300 iterations per run, a mutation rate of 0.02, a crossover rate of 0.8, and a local search probability of 0.06. The resulting binding poses were clustered based on RMSD with a 2.0 Å threshold and ranked according to their predicted binding affinities. Visual inspection and figure rendering were carried out using UCSF Chimera version 1.15 [24].

## 3. Results

### 3.1. Chemistry

The synthetic procedure to obtain target compounds **3**–**14** is summarized in Figure 2. The one-pot reaction [25] of the substituted 2-aminophenols **15**–**17** with epichlorohydrin in an aqueous basic medium gave the benzoxazine derivatives **18**–**20** (i, Figure 2, 74–84% yield). Sulfonamides **21**–**23** were obtained by reacting with *p*-toluensulfonil chloride and pyridine (ii, Figure 2, 87–89% yield). Then, the Mitsunobu reaction was performed using triphenylphosphine (Ph_3_P), diisopropylazodicarboxylate (DIAD) and the appropriate purine in anhydrous tetrahydrofuran, resulting in the formation of derivatives **3**–**5**, **7**–**9** and **11**–**13** (iii, Figure 2, 82–87% yield). Finally, the bromine on the purine heterocycle is replaced with a trifluoromethyl group, by reacting the 6-bromopurine derivatives **5**, **9** or **13**, with copper (I) iodide, methyl 2,2-difluoro-2-(fluorosulfonyl) acetate (MFSDA), and hexamethylphosphoramide (HMPA). This reaction was carried out using microwave irradiation at 150 °C for 5 min to give compounds **6**, **10** and **14** (80–86% yield).

### 3.2. In Vitro Antiproliferative Activity

The antiproliferative activity of the target compounds was evaluated using the sulforhodamine B colorimetric assay after 72 h of treatment (Table 1, Appendix A).

Compared to compound **1**, the new derivatives **3**–**14**, which feature a shortened linker and substituents on the benzoxazine ring, exhibited markedly improved antiproliferative activity. This highlights the positive impact of aromatic ring substitution. However, compared to compound **2**, which has the same substitution pattern but a longer, more flexible linker, the truncated analogues maintained their cytotoxic potency. This suggests that truncation preserves biological activity without significantly enhancing it.

The presence of a bromine substituent (**7**–**10**, IC_50_ = 4.06 µM–7.31 µM) or a methyl group (**11**–**14**, IC_50_ = 3.39 µM–7.78 µM) at the 6-position of the benzoxazine increases the antiproliferative activity against the MCF-7 breast cell line compared to chlorine at the 7-position (**3**–**6**, IC_50_ = 6.35 µM–13.60 µM). This pattern is consistent with the trend observed in HCT-116 colon cancer cells, where the substitution of a bromine (**7**–**10**, IC_50_ = 4.80 µM–8.26 µM) or a methyl group (**11**–**14**, IC_50_ = 5.20 µM–11.49 µM) at the 6-position of the benzoxazine enhances the antiproliferative activity compared to a chlorine substituent at the 7-position (**3**–**6**, IC_50_ = 6.41 µM–12.65 µM). These results suggest that electron-donating or moderately bulky substituents at the 6-position favor interaction with cellular targets, potentially by improving lipophilicity or membrane permeability.

No clear structure-activity relationships were observed for purine substitution in either breast or colon cancer cells. indicating that the benzoxazine ring plays a dominant role in modulating cytotoxicity. This is consistent with previous studies highlighting the pharmacological relevance of benzoxazine substitution patterns in determining biological activity.

Compounds **9** and **12** are the most active against MCF-7 breast cancer cells, with IC_50_ values of 4.06 and 3.39 µM, respectively. Compounds **10** and **12** are the most active against the HCT-116 tumor line, with IC_50_ values of 4.80 and 5.20 µM, respectively. Notably, compound **12**, which combines a methyl linker, a 6-methylbenzoxazine, and a dichloropurine, exhibited consistent activity across both cell lines, suggesting a favorable balance of physicochemical and pharmacophoric features. These findings support the hypothesis that strategic substitution at the benzoxazine ring, combined with linker truncation, can generate compounds with enhanced and broad-spectrum antiproliferative effects.

### 3.3. Kinase Inhibition Assays and Computational Studies

In order to study the involvement of kinase inhibition in cell death, we selected the most active compounds based on their antiproliferative activity. Specifically, compounds **5**, **8**, **9**, **10**, **11** and **12** were selected for their remarkable efficacy. These compounds were tested against a panel of 5 kinases, AMPK alpha1, HER2, ERK2, JNK1 and the LKB1/MO25/STRADa complex, that are frequently dysregulated in cancer, making them critical targets for therapeutic development. The results presented in Table 2 show residual kinase activity at two concentrations, 5 μM and 50 μM.

The compounds generally show moderate inhibition against the kinases tested. In particular, the maximum inhibition observed is against JNK1, with 60% inhibition for compounds **5**, **8**, **10** and **12**. These data also indicate that the most active compound **11** shows inhibitory activity against HER2 kinases. Specifically, **12** shows residual activities of 41% against JNK1 and 53% against HER2 at 50 μM. Interestingly and surprisingly, compound **9** exhibits no inhibition of JNK1 or HER2, with residual activity of approximately 80%.

Compared to compound **2**, which also exhibited dual inhibition of HER2 and JNK1 with residual activities of 50% and 56%, respectively, **12** exhibited a similar inhibitory profile, even displaying slightly stronger inhibition of JNK1, despite its truncated linker. This finding supports the idea that reducing the linker does not compromise kinase engagement, and may even improve synthetic accessibility.

Overall, the results highlight the potential of compound **12** as a promising candidate for further development as an anticancer agent, particularly in cancers where HER2 and JNK1 are dysregulated, such as breast and colon cancer [26].

To gain further structural insight into the kinase inhibition profile of compound **12**, molecular docking studies were performed against the ATP-binding sites of hHER2 and hJNK1. These analyses aimed to elucidate the binding mode of the compound’s enantiomers and rationalize the inhibitory activity observed experimentally.

Thus, a docking analysis of compound **12** was conducted to shed light on its molecular binding mode with hHER2 and hJNK1 kinase isozymes. Given that this compound is considered a purine analog, the ATP-binding cavity was examined as the potential inhibition site. Consistent with the criteria from our previous studies [14], we used human epidermal growth factor receptor 2 (hHER2, PDB ID: 3RCD) [27] to investigate the molecular interactions of both enantiomers of **12**.

According to the predicted docking poses of (*R*)-**12**, the ligand points the benzoxazine toward the hinge region, where it is flanked by hydrophobic residues such as Leu726, Val734, Leu800, Met801, and Leu852 on both sides of the ring. The purine is positioned in a pocket at the entrance of the catalytic site formed by residues Ser1002, Thr1003, Asp808, and Glu812. It forms a hydrogen bond with the backbone of Cys805. The phenylsulfonamide is positioned deep within the catalytic site in a hydrophobic cleft formed by Ala751, Thr798, and the alkyl chain of Lys753. In contrast, (*S*)-**12** partially aligns with its (*R*)-enantiomer, but it positions the purine and phenylsulfonamide moieties toward the solvent-exposed area in a U-shaped conformation, which removes the purine from the pocket at the entrance to the catalytic site (Figure 3). This atypical orientation of (*S*)-**12** likely underlies its reduced binding affinity, as predicted by its less favorable AutoDock score (Table 3).

The molecular conformation of compound **12** was also studied on human mitogen-activated protein kinase 8 (hJNK1, PDB ID 4AWI) [28]. In comparison to the hHER2 crystal structure (PDB ID 3RCD), the hJNK1 structure (PDB ID 4AWI) exhibits a narrower ATP-binding site entrance and lacks the pocket located at the entrance of the catalytic site. Additionally, the hJNK1 crystal structure features a larger binding site pocket that can accommodate compound **12**. Both enantiomers of **12** were docked into the ATP-binding site of the hJNK1 isoform (PDB ID: 4AWI). In the docking conformation, the purine moiety of each enantiomer is deeply inserted into the active-site cavity and adopts a mutually orthogonal orientation while residing in the sub-P-loop region. The sulfonamide functional group of both enantiomers is oriented deep into the catalytic site and toward the hinge region. A hydrogen bond is observed between the sulfonic oxygen atom of (*S*)-**12** and the backbone chain of Met111.

The primary difference in the orientation of the enantiomers lies in the positioning of their phenylsulfonamide and benzoxazine moieties. (*S*)-**12** positions its benzoxazine ring near the hinge region, close to the hydrophobic pocket formed by Val40, Ala53, Leu110, and Ile32 on one side and Leu168 on opposite side. Its phenylsulfonamide is directed toward the entrance of the catalytic site and forms a hydrogen bond with Met111. In contrast, (*R*)-**12** positions its benzoxazine ring at the entrance of the catalytic site, which is facilitated by a hydrogen bond with Asn114. Its phenylsulfonamide moiety is oriented toward the same hydrophobic region occupied by the benzoxazine ring in (*S*)-**12** (Figure 4).

The similar docking conformations of the two enantiomers on this enzyme correlate with their comparable predicted binding free energies and inhibition constants (Ki), which supports a shared binding mode (Table 3).

Molecular docking analysis revealed notable differences in the binding orientation and affinity of compound **12** enantiomers toward hHER2 and hJNK1 kinases. For hHER2, (*R*)-**12** adopted a more favorable conformation that engaged key residues within the catalytic site. This correlated with a higher predicted binding affinity. In contrast, (*S*)-**12** displayed a less optimal orientation, resulting in reduced binding affinity. However, both enantiomers exhibited similar binding modes in hJNK1, with comparable binding energies and inhibition constants.

A comparative docking analysis of compound **12** and the reference compound **2**, which differ only in linker length, revealed that they have similar binding orientations and predicted affinities in both HER2 and JNK1. Despite its reduced flexibility, compound **12** adopts poses that closely resemble those of **2**, with conserved interactions at the hinge and catalytic regions (see Appendix A for details). These findings are consistent with the comparable experimental inhibition profiles observed for both compounds, suggesting that linker truncation does not impair kinase engagement.

To further explore their pharmacological relevance, compounds **9** and **12** were selected for extended in silico and biological evaluation. These two candidates not only demonstrated strong antiproliferative activity in vitro but also differed in their structural features and kinase inhibition profiles, compound **12** acting as a dual HER2/JNK1 inhibitor, while compound **9** lacked measurable kinase activity. This mechanistic divergence provided a valuable basis to investigate how molecular properties translate into distinct pharmacokinetic behavior and modes of action.

### 3.4. In Silico Physicochemical Parameter and ADMET Predictions

#### 3.4.1. Physicochemical Parameters

The link between physicochemical properties and molecular structure is well established today. The Lipinski rules are a relevant example, indicating limits to physicochemical parameters that favor good oral absorption: the number of hydrogen bonds donors (HBD) ≤ 5, the number of hydrogen bonds acceptors (HBA) ≤ 10, molecular weight (MW) ≤ 500, and an octanol-water partition coefficient (log *p*) ≤ 5 [29]. According to Veber’s rules, molecules with ≤10 rotatable bonds (nRB) and a topological polar surface area (TPSA) ≤ 140 Å^2^ are more likely to exhibit good oral bioavailability [30]. Molar refractivity (MR) affects oral bioavailability by modulating drug permeability across membranes, interactions with transporters and metabolism, ultimately influencing absorption and distribution in the body. The MR value should be between 40 and 130 for good absorption and oral bioavailability. Acceptable MR values, in combination with the nRB, indicate adequate intestinal absorption and oral bioavailability of molecules [31].

We calculated the Lipinski and Veber rules and MR for **9** and **12** (Table 4) using the freely available website: http://www.swissadme.ch/ (accessed on 6 June 2025). The SMILE molecular structures of the compounds were obtained from PubChem (https://pubchem.ncbi.nlm.nih.gov, accessed on 6 June 2025).

Physicochemical evaluation indicates that compounds **9** and **12** exhibit favorable properties for oral bioavailability, though they slightly exceed the MW threshold defined by Lipinski’s rule (MW ≤ 500). Compound **12** (504.39 Da) marginally exceeds this limit, while compound **8** (579.26 Da) shows a more pronounced deviation. Nevertheless, both compounds comply with the other Lipinski criteria, including acceptable logP values, the absence of HBD, and six HBA, which supports good membrane permeability. Additionally, both compounds meet Veber’s criteria, with low nRB (four rotatable bonds) and moderate TPSA values (98.59 Å^2^), suggesting good oral bioavailability. Their MR values also fall within the optimal range, reinforcing their potential for adequate intestinal absorption.

#### 3.4.2. ADMET Properties

In silico ADMET evaluation have been developed as an additional tool to assist medicinal chemists in the design and optimization of leads [32]. In order to predict the pharmacokinetic properties of absorption, distribution, metabolism, and excretion, as well as the potential toxicity compounds **9** and **12** using a free website: http://biosig.unimelb.edu.au/pkcsm/prediction (accessed on 6 June 2025). These results are shown in Table 5.

The Human Intestinal Absorption (HIA) parameter is used to measure the absorption of drugs from the intestine into the circulation. It refers to the percentage of an orally administered drug that is absorbed through the human intestine and enters the systemic circulation. A high HIA (>80%) indicates good absorption and potential for effective oral administration. A low HIA (<30%) indicates poor absorption, which may require alternative formulations or routes of administration [33]. Compounds **9** and **12** show a high percentage of intestinal absorption (HIA > 95%).

Central Nervous System (CNS) permeability represented as logPS (logarithm of the permeability-surface area product), indicates the ability of a drug to cross the blood–brain barrier (BBB) and enter the CNS. Substances with logPS > −2 are considered to penetrate the CNS, while those with logPS < −3 do not penetrate the CNS [34]. These compounds have values < −3, meaning they have minimal or no ability to cross the blood–brain barrier (BBB). This property is beneficial in conventional chemotherapy for peripheral cancers (e.g., breast, lung, or colon), where BBB penetration is unnecessary, helping to prevent unwanted side effects.

CYP1A2 and CYP2D6 are cytochromes involved in drug metabolism that help prevent drug accumulation and reduce the risk of toxicity [35]. Inhibition of these enzymes may result in prolonged drug half-life, unpredictable plasma levels, and increased likelihood of adverse effects. Compounds **9** and **12** do not inhibit these cytochromes. This suggests that they are metabolized without the risk of overaccumulation.

The total clearance expressed as log (mL/min/kg), is the volume of plasma completely cleared of the drug per unit of time by the organ eliminating the drug from the body [36]. Values > 0 indicate rapid elimination; values between 0 and −1 suggest moderate elimination; and values < −1 denote slow elimination. The values obtained for **9** and **12** indicate rapid elimination.

The Ames test is a widely used assay to evaluate the mutagenic potential of drugs [32], serving as an indicator of possible carcinogenicity. Compounds **9** and **12** tested negative in the AMES toxicity, suggesting they are not mutagenic (Table 5).

The in silico ADMET evaluation of compounds **9** and **12** indicates that they have excellent intestinal absorption (>95%), making them suitable for oral administration. Their low CNS permeability (logPS < −3) minimizes the risk of unwanted neurological effects, which is advantageous for non-CNS-targeting treatments. Additionally, neither compound inhibits CYP1A2 or CYP2D6 enzymes, indicating that they are efficiently metabolized without the risks of accumulation. Their rapid clearance further supports their safe elimination. Lastly, negative Ames test results confirm that the compounds are non-mutagenic, which reinforces their potential safety profile for further medicinal research.

### 3.5. Scanning Electron Microscopy Analysis

In HCT-116 control cells, SEM analysis revealed a preserved cellular morphology characterized by intact plasma membranes, dense microvilli distribution, and firm attachment to the culture substrate (Figure 5A,B). Treatment with compound **9** led to a marked reduction in cell density and a significant loss of surface microvilli (Figure 5C,D,D’). Additional apoptotic features included cell shrinkage, prominent membrane blebbing, and the presence of numerous smooth, rounded vesicular bodies protruding from the membrane. In contrast, exposure to compound **12** induced severe plasma membrane alterations, including irregular discontinuities, large transmembrane pores (pink arrows), and frank membrane rupture (Figure 5E,F). These cells also exhibited pronounced swelling, loss of membrane integrity, and signs of cytoplasmic content release. While classical blebbing was minimal, small membrane-bound vesicles were evident.

In the MCF-7 breast cancer model both compounds induced morphological alterations of differing severity, with the effects of **12** exceeding those of **9**. While untreated control cells remained intact and confluent (Figure 5G,H), cells treated with **9** (Figure 5I,J) displayed an elongated phenotype, partial microvilli loss, and mild filopodial retraction. By contrast, **12** triggered extensive membrane rupture in larger cells, accompanied by comprehensive loss of microvilli and filopodia (Figure 5K,K’,L,L’), alongside the appearance of vesicular bodies (Figure 5L,L’).

### 3.6. Cell Cycle Assays

To further dissect the impact of compounds **9** and **12** on proliferation, we performed cell-cycle analyses to evaluate their effects on phase distribution and checkpoint regulation.

Flow cytometry-based analysis distinguished cells in G_2_/G_2_, S, and G_2_/M phases (Figure 6). In HCT-116 cells, both **9** and **12** induced a highly significant reduction in the G_1_/G_2_ fraction, with concomitant decreases in G_2_/M from 8.37% in controls to 6.67% following **9** treatment (not statistically significant) and to 5.53% with **12** (*p* < 0.01) (Figure 6A–C,G). Strikingly, S-phase cells accumulated from 18.7% in untreated cultures to 49.2% with **9** and 51.3% with **12** (both *p* < 0.001) (Figure 6A–C,G). By contrast, MCF-7 cells exhibited extensive damage and membrane rupture after treatment, preventing accurate cell-cycle profiling (Figure 6E,F,H). Such rapid lysis suggests that, in this cell line, **9** and **12** bypass classical checkpoint controls to directly engage cell-death pathways. Together, these results underscore distinct, tumor-type-specific responses to our compounds and highlight the importance of assessing both cytostatic and overt cytotoxic effects.

### 3.7. Assessment of Apoptosis and Membrane Integrity by Annexin V–FITC/PI Staining

To investigate membrane-associated events following treatment with compounds **9** and **12**, we employed the Annexin V–FITC/PI assay, which enables a functional assessment of plasma membrane integrity and asymmetry. The Annexin V–FITC/PI assay revealed markedly different membrane disruption profiles in HCT-116 and MCF-7 cells treated with compounds **9** and **12** (Figure 7).

In HCT-116 colorectal carcinoma cells (Figure 7A–C,G), **9** induced a modest but statistically significant increase in the population of cells with phosphatidylserine (PS) externalization and preserved membrane integrity (Figure 7B,G) (Annexin V^+^/PI^+^: 9.2% vs. control; *p* < 0.05), whereas treatment with **12** resulted in a strong increase in the double-positive fraction (Figure 7C,G) (Annexin V^+^/PI^+^ 71.5%), indicating advanced membrane permeabilization and loss of membrane asymmetry. These changes are consistent with a lytic form of regulated cell death involving severe compromise of the plasma membrane.

Similarly, in MCF-7 breast cancer cells, both compounds induced a pronounced increase in the Annexin V^+^/PI^+^ population (59.5% for **8** and 73.9% for **12**; Figure 7E,F,H), accompanied by a smaller increase in Annexin V^−^/PI^+^ cells, suggestive of direct necrotic damage (Figure 7H).

### 3.8. Characterization of Cell Death Pathways via Western Blot Analysis of Caspase-3, -1, and -8

To investigate the specific cell death pathways activated by compound **12**, we conducted Western blot analyses focused on the expression levels of caspase-3, caspase-8, and caspase-1 in HCT-116 colorectal carcinoma cells. These proteins were selected as key markers of distinct regulated cell death pathways, including canonical apoptosis (caspase-3), extrinsic or inflammatory apoptosis (caspase-8), and pyroptosis via inflammasome activation (caspase-1).

HCT-116 cells were exposed to compound **12** for 4 h at a concentration of 50 µM, and total protein extracts were subjected to immunoblotting (Figure 8). The analysis revealed no appreciable change in procaspase-3 levels between treated and control cells, as shown in Figure 8A and its quantification panel (Figure 8A’), suggesting no overt activation of the classical apoptotic effector caspase.

Conversely, a substantial increase in caspase-8 expression was observed in cells treated with compound **12**, as shown in Figure 8B and the corresponding densitometric analysis (Figure 8B’). Additionally, a moderate but consistent elevation in caspase-1 levels was detected in treated cells compared to untreated controls (Figure 8C,C’). The signal intensity indicated an intermediate level of caspase-1 induction, suggesting a potential involvement of inflammasome-related processes.

Together, these data point to a selective pattern of caspase expression changes in response to compound **12**, marked by increased levels of caspase-8 and caspase-1, in the absence of caspase-3 activation.

## 4. Discussion

We have developed an efficient, streamlined synthetic approach to access a new series of benzoxazine–purine derivatives. This strategy incorporates diverse substituents while avoiding flexible linkers. This enables us to generate structurally compact analogues with a higher yield than our previously synthesized derivatives with an ethyl linker. [18,19]

The antiproliferative activity of compounds **3**–**14** was initially assessed in the human breast cancer cell line MCF-7 and the human colon cancer cell line HCT-116. Structure-activity relationships findings suggest that the presence of specific substituents on the benzoxazine moiety significantly influences antiproliferative activity. Modifications at the 6-position of the benzoxazine ring, particularly the presence of bromine or methyl substituents, enhance antiproliferative activity more than chlorine at the 7-position across breast and colon cancer cell lines. This consistent trend across two distinct tumor models underscores the importance of electronic and steric effects in modulating biological activity at this position. However, variations in the purine moiety did not yield a clear structure-activity relationships, suggesting that substitutions on the benzoxazine scaffold may be more critical in determining cytotoxic potency.

The antiproliferative data clearly demonstrate that shortening the linker between the benzoxazine and purine moieties preserves biological activity. In our previous work [19], compound **2**, with a longer ethyl linker, showed potent cytotoxicity against both the MCF-7 and HCT-116 cell lines. In the present study, compound **12**, which shares the same substitution pattern but features a truncated methyl linker, maintained comparable activity (IC_50_ = 3.39 µM for MCF-7 and 5.20 µM for HCT-116), suggesting that reduced conformational flexibility does not compromise efficacy.

To investigate the potential role of kinase modulation in the observed cytotoxicity, kinase profiling was conducted on the most active compounds. Compound **12** exhibited dual inhibition of JNK1 and HER2 (residual activities of 41% and 53%, respectively), whereas compound **9** lacked activity against both kinases. The experimental inhibition values of compound **12** are consistent with the docking results and can be explained by the predicted binding modes. Although (*R*)-**12** exhibited a stronger predicted binding affinity for hHER2, significant conformational differences between the enantiomers may reduce the overall inhibitory effect due to the presence of the less favorable (*S*)-**12**. In contrast, both enantiomers exhibited similar, favorable binding conformations in hJNK1, likely contributing to the higher overall inhibition observed experimentally.

Furthermore, the comparison of the docking studies of **2** and **12** revealed that both compounds adopt similar binding orientations within the HER2 and JNK1 kinase pockets. The purine ring was found to anchor the interaction, while the benzoxazine substituents were observed to contribute to hydrophobic contacts. These results suggest that the key pharmacophoric elements are conserved in both series and that linker length primarily affects molecular rigidity and spatial orientation rather than target engagement.

Compounds **9** and **12** have favorable physicochemical profiles, making them good candidates for oral administration. In silico ADMET analysis confirmed that both compounds possess excellent oral bioavailability, low neurotoxicity risk, favorable metabolic and clearance profiles, and no mutagenic potential highlighting their suitability for further therapeutic development.

The ultrastructural alterations observed by SEM support the hypothesis that **12** induces a lytic form of regulated cell death distinct from classical apoptosis. Unlike the membrane blebbing and vesiculation typically associated with apoptosis [37], treatment with **12** resulted in extensive plasma membrane disruption, including pore formation and cytoplasmic leakage-features consistent with pyroptosis, a form of inflammatory programmed cell death characterized by membrane rupture and release of intracellular content [38]. The more modest effects of **9**, including cell shrinkage and blebbing without full membrane rupture, further underscore the mechanistic divergence between the two compounds.

In line with these morphological findings, flow cytometry-based cell-cycle analysis revealed that both compounds induced a pronounced S-phase arrest in HCT-116 cells, suggesting replication stress or DNA damage that activates intra-S-phase checkpoint responses. This pattern mirrors the effects of classical genotoxic agents such as 5-fluorouracil and topoisomerase inhibitors, which interfere with DNA synthesis and can lead to apoptosis or mitotic catastrophe if damage persists [39,40]. Notably, the extent of membrane damage in MCF-7 cells treated with **9** or **12** precluded reliable cell-cycle profiling, consistent with rapid, checkpoint-independent cell death in this model. Taken together, these findings indicate that while both compounds elicit DNA damage responses, **12** more effectively promotes progression toward inflammatory, lytic cell death highlighting distinct, tumor-type-specific modes of action.

This interpretation is further supported by Annexin V–FITC/PI staining, a sensitive and versatile tool for monitoring cell fate in response to compounds that disrupt membrane homeostasis. This assay functionally corroborates the morphological evidence of plasma membrane disruption by assessing membrane asymmetry and integrity, two key hallmarks traditionally associated with apoptosis but also relevant to various forms of regulated cell death, including necrosis, pyroptosis, and other lytic mechanisms [38,41]. The observed differences between cell lines and between compounds suggest that membrane destabilization is a central event in the cytotoxic response, likely reflecting convergence of multiple regulated cell death pathways involving terminal plasma membrane permeabilization [42,43,44].

To further explore the molecular pathways underlying these distinct cell death phenotypes, we examined the activation of key caspases involved in apoptotic and inflammatory responses. Although both **9** and **12** exhibited antiproliferative activity, **12** is most potent and triggered the most prominent morphological alterations and a marked accumulation of cells in the S phase, whereas **9** induced comparatively milder effects. HCT-116 was chosen as the preferred model for mechanistic investigations due to its demonstrated functional stability during prolonged treatments [45], alongside its well-characterized molecular profile, including wild-type p53 status and robust caspase-3 expression. This combination enables comprehensive analysis of both intrinsic apoptotic pathways and alternative regulated cell death mechanisms [46,47,48,49]. In contrast, MCF-7 cells displayed excessive sensitivity to treatment, undergoing extensive lysis that impeded reliable protein extraction, further justifying the choice of HCT-116 for this analysis.

To investigate the potential cell death mechanisms triggered by **12**, we focused on the expression of caspase-3, caspase-8, and caspase-1 by Western blot. Caspase-3 is a well-established effector of canonical apoptosis, whereas caspase-8 functions not only as an initiator of apoptotic signaling but also as a key regulator of alternative cell death programs, including necroptosis and pyroptosis [50]. Caspase-1, in turn, is a central mediator of pyroptosis via inflammasome activation [51,52]. Western blot analysis performed on HCT-116 cells treated with compound **11** revealed a distinct molecular profile indicative of the activation of regulated cell death pathways, showing features divergent from classical intrinsic or extrinsic apoptosis. Interestingly, no significant changes in procaspase-3 levels between treated and untreated cells despite its role as a key executioner protease responsible for the morphological and nuclear features of classical apoptosis, including DNA fragmentation [53]. These findings suggest that caspase-3-dependent apoptosis is not the predominant cell death mechanism triggered under these conditions. In contrast, a marked increase in caspase-8 expression was detected. The prominent activation of caspase-8, in the absence of caspase-3 involvement, points toward a non-apoptotic, caspase-8-dependent form of inflammatory cell death, consistent with non-canonical pyroptosis.

Caspase-1 is a key initiator caspase in the canonical inflammasome pathway and is essential for the execution of pyroptosis, a form of inflammatory cell death characterized by gasdermin D cleavage, pore formation in the plasma membrane, cell swelling, and lysis. Although traditionally associated with immune cells, growing evidence indicates that epithelial and tumor cells can also activate caspase-1 in response to genotoxic or metabolic stress, particularly in the presence of reactive oxygen species (ROS) or endoplasmic reticulum (ER) stress [54,55]. The induction of caspase-1 in HCT-116 suggests that **12** may trigger an inflammatory response that contributes not only to direct cytotoxicity but also to a potential immunogenic effect within the tumor microenvironment.

Classical apoptosis is typically driven by caspase-3 activation; thus, its unchanged expression suggests that compound 11 triggers alternative forms of regulated cell death. Recent studies [52] support caspase-8 as a key regulator not only of apoptosis but also of pyroptosis and inflammatory death pathways. Inhibition of TAK1 has been shown to activate caspase-8, leading to gasdermin D cleavage and induction of pyroptotic cell death via membrane pore formation and inflammation [56]. Additionally, caspase-8 has been found to directly cleave gasdermin D during bacterial infection, promoting pyroptosis independently of caspase-3 [57]. These mechanistic insights align with the pronounced membrane damage and vesiculation observed by SEM and Annexin V assays in our study. Furthermore, recent reviews have emphasized the role of caspase-8 in bridging apoptotic and inflammatory signaling pathways [58]. The concomitant upregulation of caspase-1, an inflammasome effector central to pyroptosis, reinforces the hypothesis that compound **12** induces a pyroptotic or related lytic form of cell death rather than classical apoptosis.

Taken together, these data suggest that compound **12** induces a caspase-8-dependent form of inflammatory cell death, characterized by plasma membrane permeabilization and morphological features that diverge from classical caspase-3–mediated apoptosis. This interpretation is supported by the absence of caspase-3 activation, the upregulation of caspase-8 and caspase-1, and the membrane pore formation observed in treated cells. Furthermore, the kinase inhibition profile of compound **12**, notably its dual targeting of HER2 and JNK1, two kinases involved in oncogenic and stress-responsive signaling, suggests a mechanistic link between survival pathway disruption and the induction of non-apoptotic regulated cell death. Although direct evidence implicating JNK1 as a primary driver of pyroptosis remains limited, stress-activated signaling cascades involving JNK have been shown to modulate inflammasome activity and caspase-1 activation, providing a plausible connection to lytic, inflammatory modes of cell death [59].

Nonetheless, the evaluation of caspase activation alone offers only a partial view of the underlying cell death mechanism, particularly in the context of lytic or mixed phenotypes, where multiple pathways may converge. For instance, although gasdermin D (GSDMD) cleavage is a well-established hallmark of pyroptosis, its activation downstream of caspase-1 cannot be conclusively inferred from caspase expression or cleavage patterns alone. Therefore, the present findings should be regarded as a first exploratory step toward elucidating the molecular basis of compound-induced cytotoxicity. To delineate the precise mode of regulated cell death, future studies should incorporate direct assessment of gasdermin family members, additional effector molecules, and upstream regulatory nodes across multiple cell death programs. These insights lay the groundwork for further exploration of benzoxazine–purine hybrids as inducers of non-apoptotic cell death, with potential applications in resistant or inflammation-associated cancers.

## 5. Conclusions

We have developed an efficient synthetic approach to access a new series of benzoxazine–purine derivatives bearing diverse substituents and reduced conformational flexibility. These compounds exhibited low micromolar antiproliferative activity in breast and colon cancer cell models and showed favorable in silico physicochemical and ADMET parameters. In terms of mechanism of action, compound **12** acts as a dual HER2/JNK1 kinase inhibitor and induces caspase-8-dependent pyroptosis-like cell death, while compound **9** promotes a kinase-independent apoptosis-like pathway. These findings demonstrate that slight structural modifications can selectively engage distinct cell death programs. Moreover, these results align with our previous reports of purine–benzoxazine conjugates exhibiting cytotoxic activity against tumor cell lines, reinforcing the therapeutic relevance of these chemical structures. Overall, this work highlights the value of combining scaffold-based design with mechanistic profiling to develop multifunctional agents for targeted cancer therapy. Future efforts should focus on identifying downstream molecular effectors and validating efficacy in relevant in vivo models.

## Figures and Tables

**Figure 1 pharmaceutics-17-01260-f001:**
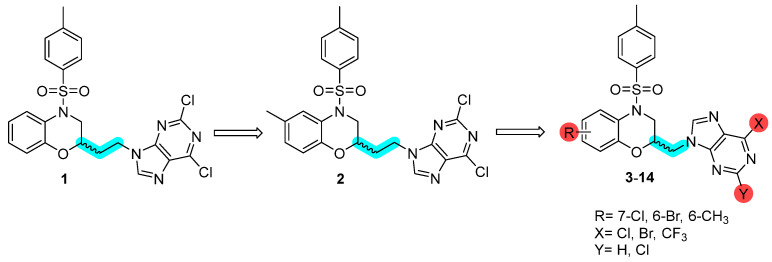
Substituted analogues from compounds **1** and **2** with a reduced number of rotatable bonds (highlighted in blue).

**Figure 2 pharmaceutics-17-01260-f002:**
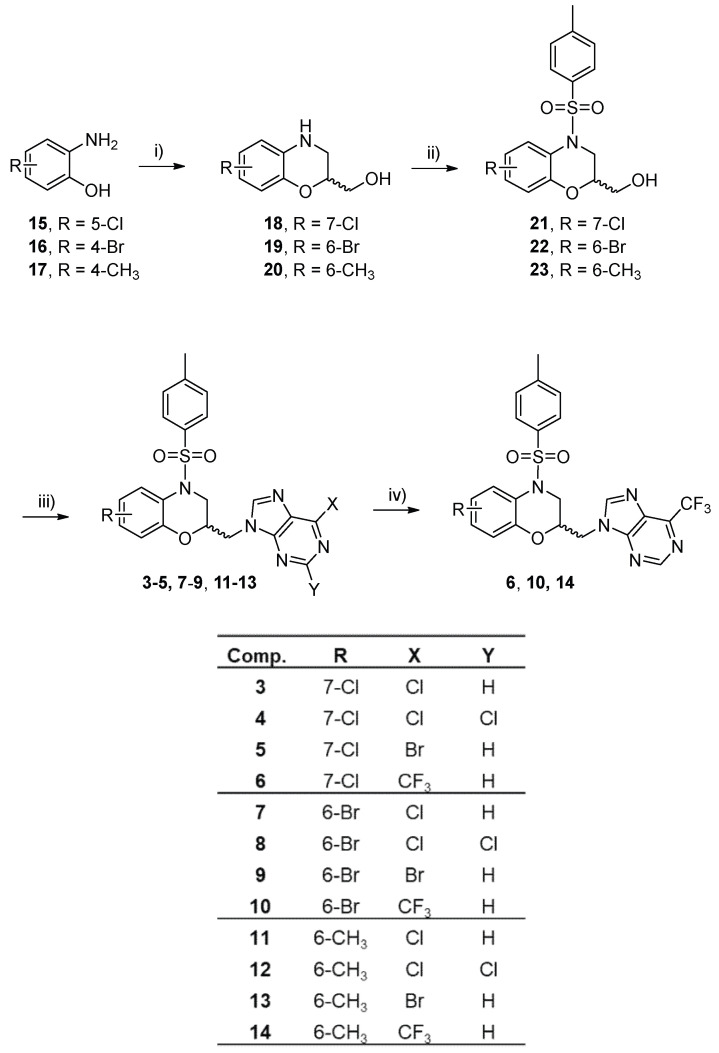
Reagents and conditions: i) epichlorohydrin, NaOH, H_2_O; ii) TsCl, pyr, CH_2_Cl_2,_ 12 h, 0 °C to rt; iii) 6-halopurine or 2,6-dihalopurine, DIAD, Ph_3_P, anhydrous THF, 48 h, −20 °C to rt; iv) **5**, **9** or **13**, MFSDA, CuI, HMAP, DMF, microwave 5 min, 150 °C.

**Figure 3 pharmaceutics-17-01260-f003:**
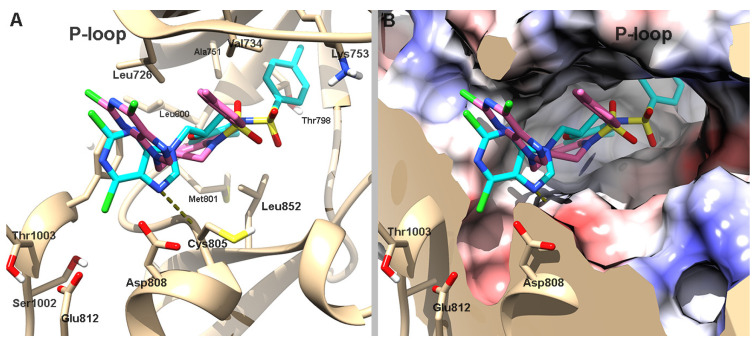
(**A**) Predicted binding orientations of (*R*)-**12** (cyan) and (*S*)-**12** (pink) within the hHER2 binding site (PDB ID: 3RCD; protein shown in tan). (**B**) Coulombic surface representation of hHER2 (white: neutral; blue: positive; red: negative electrostatic potential) with the docked poses of both enantiomers. Dashed lines indicate predicted hydrogen bond interactions.

**Figure 4 pharmaceutics-17-01260-f004:**
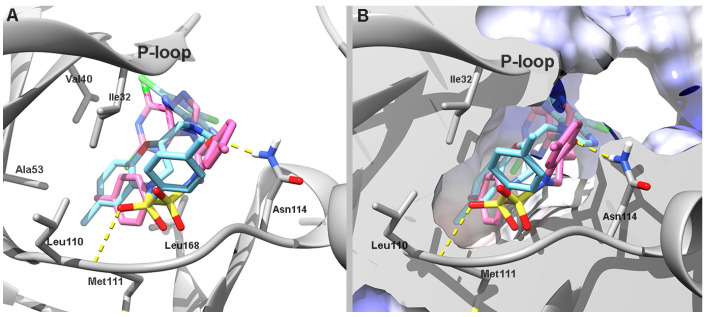
(**A**) Predicted binding orientations of (*R*)-**12** (cyan) and (*S*)-**12** (pink) within the hJNK1 active site (PDB ID: 4AWI; protein shown in grey). (**B**) Coulombic surface potential of hJNK1 (white: neutral; blue: positive; red: negative) displaying the docked poses of both enantiomers. Predicted hydrogen bonds are shown as dashed yellow lines.

**Figure 5 pharmaceutics-17-01260-f005:**
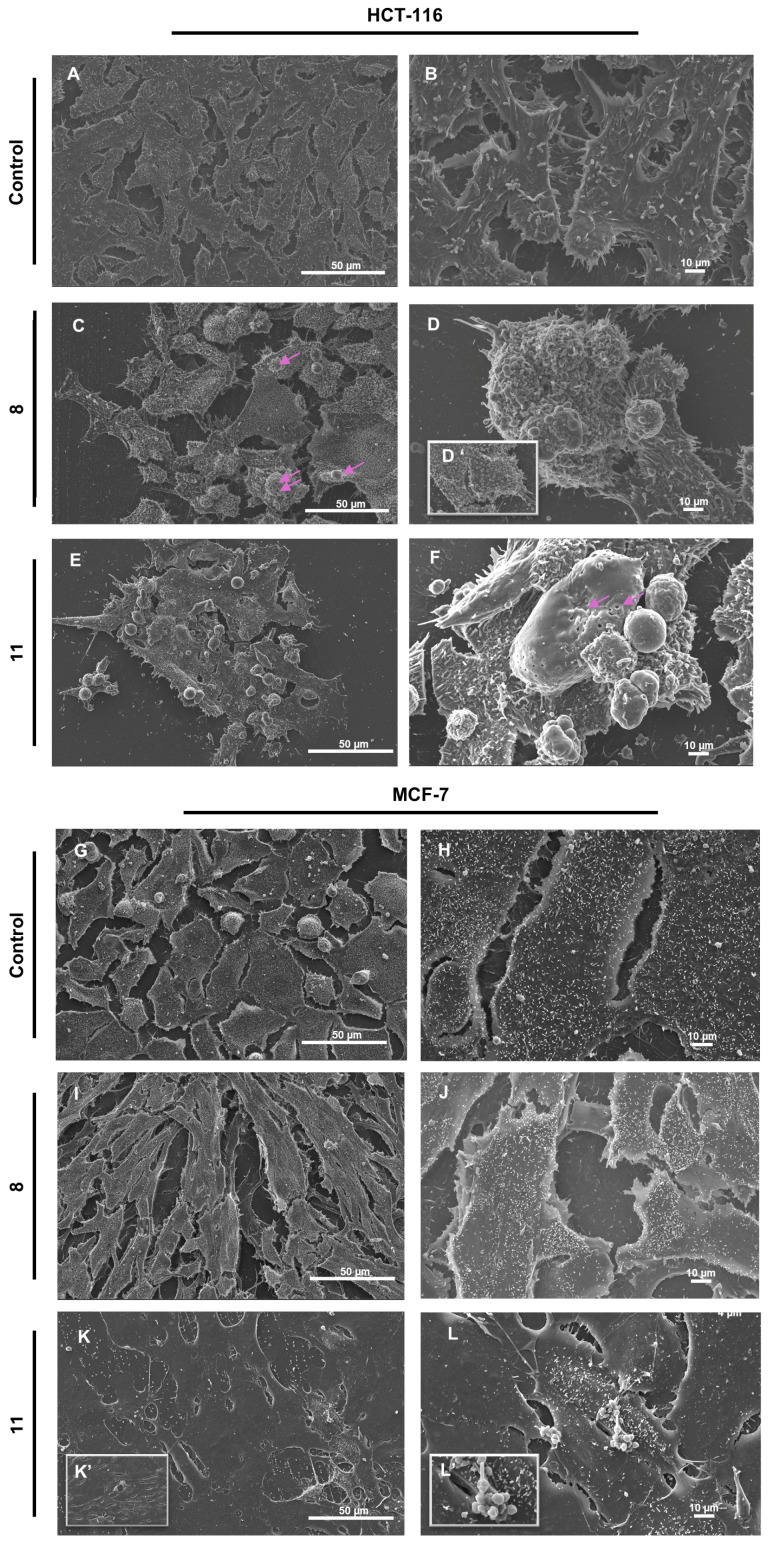
Scanning electron microscopy of HCT-116 and MCF-7 cells treated by **9** (**C**,**D**,**D’**,**I**,**J**), **12** (**E**,**F**,**K**,**K’**,**L**,**L’**) and control (**A**,**B**,**G**,**H**). Control HCT-116 cells displayed preserved membrane integrity, dense microvilli, and strong adherence to the substrate (**A**,**B**). Treatment with **9** induced apoptotic-like features, including reduced cell density, microvilli loss, membrane blebbing (indicated by the pink arrows), and vesicle formation without complete membrane rupture (**C**,**D**,**D’**). In contrast, **12** caused extensive plasma membrane damage, with visible discontinuities, large transmembrane pores (indicated by the pink arrows), cell swelling, and cytoplasmic leakage, consistent with pyroptosis (**E**,**F**). In MCF-7 cells, **9** caused moderate membrane changes such as elongation, partial microvilli loss, and filopodial retraction (**I**,**J**), whereas **12** led to severe membrane rupture, complete loss of surface structures, and vesicle formation (**K**,**K’**,**L**,**L’**). Control MCF-7 cells remained intact and confluent (**G**,**H**). Scale bars as indicated.

**Figure 6 pharmaceutics-17-01260-f006:**
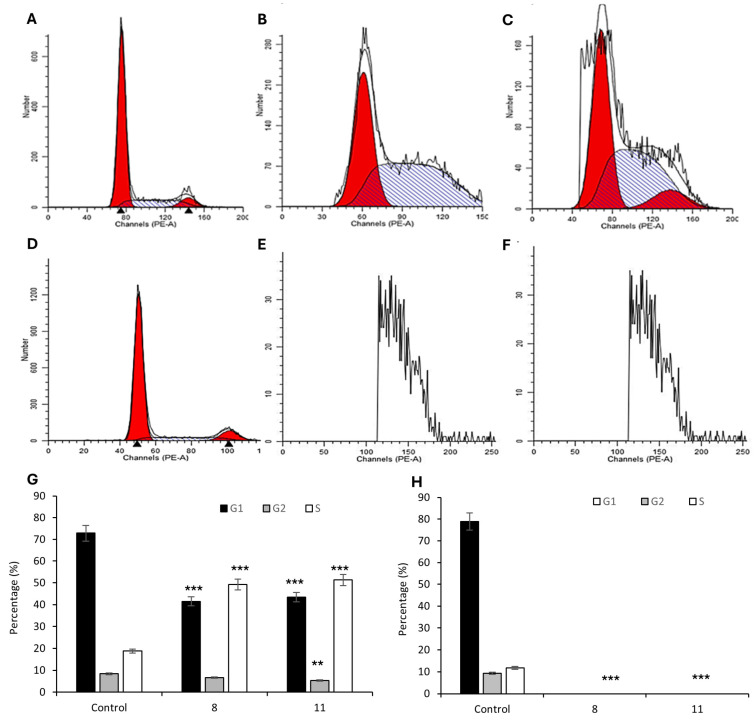
Cell cycle of HCT-116 colon cancer cells and MCF-7 breast cancer cells treated by **9** (**B**,**E**) and **12** (**C**,**F**) *vs*. control (**A**,**D**). Percentage of HCT-116 (**G**) and MCF-7 (**H**) in the cell cycle. Data are expressed as the mean ± SD of the mean of three independent experiments (** *p* < 0.01 vs. control and *** *p* < 0.001 vs. control). The red areas correspond to the G_0_/G_1_ phase, the grey areas represent the S phase, and the black areas indicate the G_2_/M phase populations.

**Figure 7 pharmaceutics-17-01260-f007:**
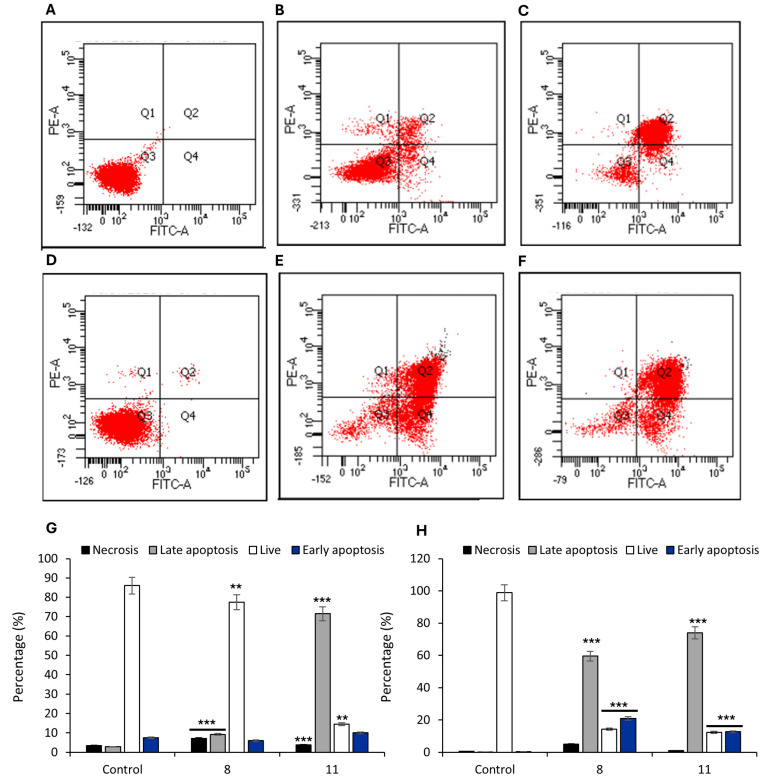
Apoptosis analysis of HCT-116 colon cancer cells and MCF-7 breast cancer cells treated with compounds **9** and **12**. Representative flow cytometry plots showing apoptosis in HCT-116 cells treated with **9** (**B**), **12** (**C**), and untreated control (**A**); and in MCF-7 cells treated with **9** (**E**), **12** (**F**), and untreated control (**D**). Quantification of cell populations ((**G**) for HCT-116 and (**H**) for MCF-7) is expressed as the percentage of necrotic, late apoptotic, early apoptotic, and live cells based on Annexin V-FITC and PI staining. Data represent mean ± SD from at least three independent experiments. Statistical significance: ** *p* < 0.01, *** *p* < 0.001 vs. control.

**Figure 8 pharmaceutics-17-01260-f008:**
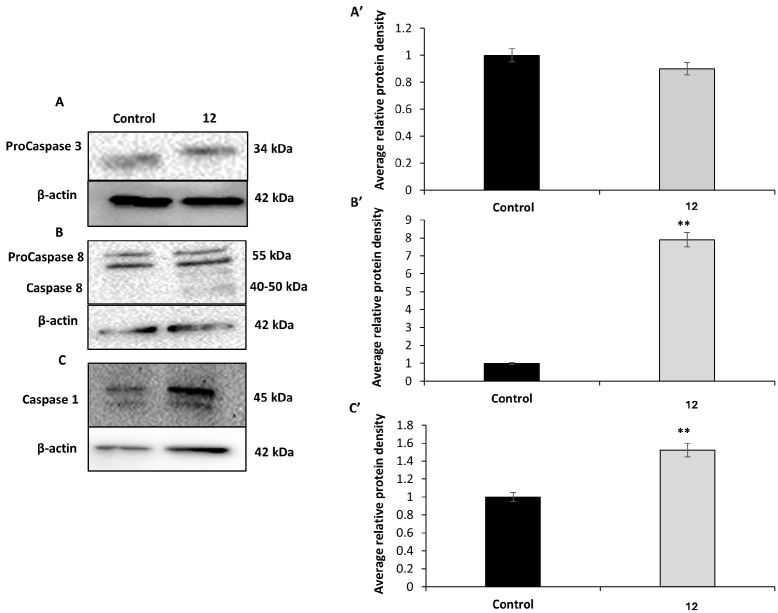
Western blot analysis of caspase activation (caspases-3, -8, and -1) as an initial approach to elucidate the regulated cell death mechanisms triggered by compound **12** in HCT-116 cells. Protein bands from (**A**) pro-caspase-3, (**B**) pro-caspase and caspase-8, (**C**) caspase-1. Relative quantification of proteins of (**A’**) pro-caspase-3, **(B’**) pro-caspase and caspase-8, (**C’**) caspase-1. Western Relative quantification of proteins normalized with β- actin signal and relative to control cells (value 1). Statistical significance: ** *p* < 0.01 vs. control.

**Table 1 pharmaceutics-17-01260-t001:** Antiproliferative activities of compounds **3**–**14** against MCF-7 and HCT-116 cancer cells. Compounds **1** and **2** are included for reference.

Comp.	R	X	Y	IC_50_ MCF-7 (μM)	IC_50_ HCT-116 (μM)
**3**	7-Cl	Cl	H	12.84 ± 0.76	6.41 ± 0.52
**4**	7-Cl	Cl	Cl	11.96 ± 0.37	11.56 ± 0.42
**5**	7-Cl	Br	H	6.35 ± 0.21	6.52 ± 0.05
**6**	7-Cl	CF_3_	H	13.60 ± 0.59	12.65 ± 0.34
**7**	6-Br	Cl	H	7.31 ± 0.54	7.80 ± 0.42
**8**	6-Br	Cl	Cl	5.74 ± 0.09	8.26 ± 0.57
**9**	6-Br	Br	H	4.06 ± 0.16	7.07 ± 0.21
**10**	6-Br	CF_3_	H	4.50 ± 0.84	4.80 ± 0.18
**11**	6-CH_3_	Cl	H	6.70 ± 1.57	7.32 ± 0.25
**12**	6-CH_3_	Cl	Cl	3.39 ± 0.079	5.20 ± 0.23
**13**	6-CH_3_	Br	H	7.78 ± 0.28	11.49 ± 0.32
**14**	6-CH_3_	CF_3_	H	7.51 ± 0.75	7.85 ± 0.35
**1** [18]	H	Cl	Cl	13.00 ± 0.11	7.06 ± 0.80
**2** [19]	6-CH_3_	Cl	Cl	2.27 ± 0.61	4.44 ± 0.33
**5-FU** [18]	-	-	-	1.5 ± 0.31	2.40 ± 0.62

All experiments were conducted in duplicate and gave similar results. Data are mean ± SD of three independent determinations. MCF-7: human breast cancer cell line and HCT-116: human colon cancer cell line.

**Table 2 pharmaceutics-17-01260-t002:** Selective profiles of **5**, **8**, **9**, **10**, **11** and **12** against 5 protein kinases. Compound **2** [19] is included for reference.

Kinase	Concentration	5	8	9	10	11	12	2 [19]
AMPK alpha1	5 μM50 μM	81	86	88	82	86	88	93
58	75	88	84	85	74	75
HER2	5 μM50 μM	77	59	73	81	82	76	90
67	63	81	63	92	53	50
ERK2	5 μM50 μM	81	77	80	81	82	81	88
70	64	83	73	81	72	73
JNK1	5 μM50 μM	75	57	73	68	66	77	96
41	42	82	40	61	41	56
LKB1/MO25/STRADa	5 μM50 μM	86	82	83	82	78	82	88
100	84	92	97	104	98	89

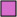
 Residual activity < 55%; 
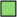
 Residual activity ≥ 55% and ≤ 75%; 
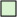
 Residual activity > 75%.

**Table 3 pharmaceutics-17-01260-t003:** Predicted binding energy and inhibition constant calculated with Autodock 4.2.6 of **12** on hHER2 (PDB ID: 3RCD) and hJNK1 (PDB ID: 4AWI), based on the calculated binding poses.

Comp.	*h*HER2		*h*JNK1	
	ΔG (Kcal/mol)	*K*_i_ (*n*M)	ΔG (Kcal/mol)	*K*_i_ (*n*M)
(*R*)-**12**	−9.98	48.49	−9.81	64.38
(*S*)-**12**	−8.53	563.73	−9.95	50.65

**Table 4 pharmaceutics-17-01260-t004:** Calculated Lipinski and Veber parameters for compounds **9** and **12**.

Comp.	HBD	HBA	MW	LogP	nVs	nRB	TPSA	MR
Lipinski *	≤5	≤10	≤500	≤5	-	-	-	
Veber **	-	-	-		-	≤10	≤140	
**9**	0	6	579.26	3.54	1	4	98.59	130.61
**12**	0	6	504.39	3.87	-	4	98.59	130.19

* Lipinski reference values; ** Veber reference values; HBD, number of hydrogen bond donors; HBA, number of hydrogen bond acceptors; MW, molecular weight; LogP, lipophilicity (*O*/*W*); nVs, number of Lipinski rule violations; nRB, number of rotatable bonds; TPSA, topological polar surface area (TPSA) (Å^2^); MR, molar refractivity.

**Table 5 pharmaceutics-17-01260-t005:** ADMET properties of compounds **8** and **11**.

Comp.	HIA (%)	CNS Permeability (logPS)	CYP1A2 Inhibition	CYP2D6 Inhibition	Total Clearance (log mL/min/kg)	AMES Toxicity
**9**	95.54	−3.40	No	No	0.04	No
**12**	95.58	−3.36	No	No	0.07	No

HIA, Human Intestinal Absorption (%); CNS Permeability (logPS); Cytochrome CYP1A2; Cytochrome CYP2D6; Total clearance (log mL/min/kg).

## Data Availability

The data presented in this study are available in this article and Appendix A. Further inquiries can be directed to the corresponding author(s).

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
