# Peer review of "Benzoxazine–Purine Hybrids as Antiproliferative Agents: Rational Design and Divergent Mechanisms of Action"

_pharmaceutics, 2025, doi:10.3390/pharmaceutics17101260_

Round 1

Reviewer 1 Report

Comments and Suggestions for Authors

This manuscript describes the “Targeted cancer therapy with benzoxazine–purine hybrids: Rational design and divergent mechanisms of action". The authors established a novel approach to in vitro anti-proliferative activity, rational design of benzoxazine-purine hybrids, resulting in the discovery of compounds with distinct mechanisms of action. This reported protocol could interest the readers; therefore, a minor revision is acceptable to this journal.
Specific comments
1)
I recommend consulting a few recent articles in the field “Benzoxazine–Purine Hybrids” to ensure the work is situated within the broader context of existing research. This will help in strengthening the rationale for your study.
2)
Modified the conclusion based on research findings rather than general statements. Please ensure that your conclusions section emphasizes the scientific value added by your paper and the applicability of your findings. Highlight the novelty of your study. In addition to summarising the actions taken and the results, please explain their significance in the conclusions.
3)
A brief description of the method of synthesis and characterization of the new compounds should be added in the abstract.
4)
What is the source of these 6-halopurine or 2,6-dihalopurine compounds, and should provide the preparation method.
5)
The authors cited ‘Scheme 1’ in the main manuscript, but did not provide the ‘Scheme 1’ data.
6)
What is the reason the authors used a wavy bond in Figure 2?
7)
Authors have performed typical docking studies. In my opinion, it is better to launch molecular dynamics simulations to validate the molecular docking study and to confirm the stability of the obtained compounds.
8)
Please outline a comprehensive section to discuss the structure-activity relationship and favoured key structural features (pharmacophores) and how far they are influencing the activities under investigation based on experimental and computational studies.
9)
Why did you only perform ADMET properties on 2 molecules and not the full set? Please provide the full set of calculations. Those are small molecules and should be easy to perform.
10)
In the supporting information, the integration values shown in the 1H NMR spectra of compound 17 do not look reasonable. The proton signal assigned to the CH2OH group of 17 in its NMR does not show the right integration. I suggest that the authors further purify these compounds and revise the yields accordingly.

Reviewer 2 Report

Comments and Suggestions for Authors
  1. Please add the HRMS spectra within the Supplementary materials
  2. Line 127 "The corresponding substituted 2-aminophenol (14, 15, 16) 6 mmol is dissolved in 30 mL of H2O in presence of 8.4 mmol of NaOH." sentences should be in past tense as the experiment has already be completed. (see the entire experimental work)
  3. No any experimental drug (positive reference) was taken for comparison of cytotoxicity results
  4. Modify the title as the compound is an experimental one (not for cancer therapy, its a preliminary report).
  5. Overall a nice piece of research work
Comments on the Quality of English Language

Somewhere proof reading is required especially.

Reviewer 3 Report

Comments and Suggestions for Authors

The present manuscript describes the synthesis of a series of novel benzoxazine-purines and their estimation as antiproliferative agents acting through different mechanisms of cell death. The target compounds were obtained via a straightforward synthetic method with good yields and their antiproliferative and related properties investigated with a proper set of methods. Peculiar is the change of mechanism of action of the compounds due to modest structure modification.

A similar work (https://doi.org/10.1016/j.bmc.2024.117849) of the authors has been published reporting the compounds differing from the currently obtained ones by a CH2-group. Unfortunately, this work is not cited in the introduction or discussed in any way. It is cited in the middle of the text (ref. 34) relating to a brief comment on the synthesis and not at all reflecting the importance of this publication for the context of the work and estimation of its novelty.

Though I suppose, the present manuscript nevertheless can be published, I would like to recommend being more accurate with reporting the investigation background.

Also, the following remarks should be addressed.

– It would be more reasonable to compare the obtained benzoxazine-purines not to compound 1, but to the series of compounds obtained in the authors’ previous publication [34]. These two sets of compounds differ only by the linker length, and their comparison could help to conclude, how the change of linker, proposed in the current work, influences biological activity. The comparison of the binding modes would also be of interest.

– Data for a reference compound in description of MTT test are absent.

– As the purity of the target compounds is confirmed by 1H NMR spectra (no HPLC or combustion analysis data are given), the spectra images should be given in the full range of chemical shifts starting from 0 ppm. According to 13C NMR spectra of compounds 3, 4, 5, 6, 9, purity of these compounds is questionable.

19F NMR spectra should be registered for CF3-derivatives. As for none of CF3-containing compounds I managed to find quartets corresponding to CF3-groups (for compound 9 it may be at 120.5 ppm, not at 123.0 as described, but it is dubious), the presence of fluorine in molecules requires better evidences.

– In description of 13C NMR spectra chemical shifts should be given to one decimal place (152.1 not 152.08 ppm).

– For previously reported compounds 18,19 references should be given in Supplementary, alongside each compound description. 

Round 2

Reviewer 2 Report

Comments and Suggestions for Authors

Manuscript is ok now

Reviewer 3 Report

Comments and Suggestions for Authors

In general, the authors took into account the comments of the reviewer in the revised manuscript, so the manuscript can be published in present form.